# LI-Detector: a Method for Curating Ordered Gene-Replacement Libraries

Emily C. A. Goodall,[a] Faye C. Morris,[b] Samantha A. McKeand,[b] Rudi Sullivan,[b] Isabel A. Warner,[a] Emma Sheehan,[b] Gabriela Boelter,[b] Christopher Icke,[a] Adam F. Cunningham,[c] Jeffrey A. Cole,[b] Manuel Banzhaf,[b] Jack A. Bryant,[b] Ian R. Henderson[a]

[a]Institute for Molecular Bioscience, University of Queensland, Brisbane, Queensland, Australia
[b]Institute of Microbiology and Infection, University of Birmingham, Birmingham, United Kingdom
[c]Institute of Immunology and Immunotherapy, University of Birmingham, Birmingham, United Kingdom

**ABSTRACT** In recent years the availability of genome sequence information has grown logarithmically resulting in the identification of a plethora of uncharacterized genes. To address this gap in functional annotation, many high-throughput screens have been devised to uncover novel gene functions. Gene-replacement libraries are one such tool that can be screened in a high-throughput way to link genotype and phenotype and are key community resources. However, for a phenotype to be attributed to a specific gene, there needs to be confidence in the genotype. Construction of large libraries can be laborious and occasionally errors will arise. Here, we present a rapid and accurate method for the validation of any ordered library where a gene has been replaced or disrupted by a uniform linear insertion (LI). We applied our method (LI-detector) to the well-known Keio library of *Escherichia coli* gene-deletion mutants. Our method identified 3,718 constructed mutants out of a total of 3,728 confirmed isolates, with a success rate of 99.7% for identifying the correct kanamycin cassette position. This data set provides a benchmark for the purity of the Keio mutants and a screening method for mapping the position of any linear insertion, such as an antibiotic resistance cassette in any ordered library.

**IMPORTANCE** The construction of ordered gene replacement libraries requires significant investment of time and resources to create a valuable community resource. During construction, technical errors may result in a limited number of incorrect mutants being made. Such mutants may confound the output of subsequent experiments. Here, using the remarkable *E. coli* Keio knockout library, we describe a method to rapidly validate the construction of every mutant.

**KEYWORDS** *E. coli*, gene-knockout, Keio, library validation, sequencing, TraDIS, Tn-Seq, knockout, mutant, transposon insertion sequencing

Advances in whole genome sequencing have led to an exponential increase in the number of sequenced genomes, and consequently, an increase in the number of predicted genes. However, the functional annotations of these genes are lagging. Functional annotations are a fundamental resource to the field of biology; uncharacterized genes may encode enzymes of biotechnological importance, disease determinants, and antimicrobial targets, and may offer information about strain speciation, evolution, and diversity. Hence, to harness these untapped resources, it is important to develop systematic approaches to increase functional annotations of genes.

In recent decades there has been significant improvement in high-throughput approaches that can assay thousands of mutants to link phenotype to genotype (1). The mutant libraries used in these screens generally fall into one of two categories: pooled libraries or ordered libraries. A pooled library comprises a heterogeneous collection of mutants, whereas an ordered library is a collection of arrayed mutants where

Address correspondence to Ian R. Henderson, i.henderson@uq.edu.au.

The authors declare no conflict of interest.

each individual mutant is stocked independently of others (2). Screening pooled libraries can offer a much higher level of genetic resolution (3). However, the number of mutants in such libraries can be limiting in experiments with a tight bottleneck (4). Smaller, defined libraries can therefore be preferential in relevant animal studies. An additional strength of screening ordered libraries is the ability to assay each mutant independently and to access individual mutants for further study. Ordered libraries now exist for several diverse organisms, and these collections serve as a fundamental resource for the research community to conduct further functional assays (1, 5–13). However, in smaller ordered libraries each mutant is often represented by a small number of independent copies (typically 1–2); this places greater importance on each individual mutant being correct.

While ordered gene-deletion libraries are excellent resources for linking phenotype to genotype, errors during the construction of gene-deletion mutants can arise, such as gene-duplication events, secondary suppressor mutations, mistargeting of the selection cassette, or accidental interruption of adjacent genes (14, 15). In addition, library stocking errors and neighboring well contamination can arise when creating or using such libraries (11, 12). As such, it is important to verify the accuracy of each mutant when using such libraries. Whole-genome sequencing of each individual mutant would prove challenging and costly, thus previous attempts to validate ordered libraries have relied on PCR to check the site of insertion of the selection cassette (14). While such analyses are useful, they do not define the precise point of insertion or whether an adjacent coding sequence is interrupted. Furthermore, if the amplicons for the mutant and wild-type (WT) bacterium are of a similar size it can be difficult to discern if the mutant was constructed correctly. Thus, a method to rapidly validate mutations within gene-deletion libraries would be useful.

Recently, we benchmarked TraDIS (Transposon-Directed Insertion-site Sequencing) (16), a Transposon Insertion Sequencing (TIS) (17) method that makes use of a pooled library of insertion mutants against an ordered library of gene-deletion mutants within the same strain, the Keio library, as a method of defining essential genes (3, 13). After detailed analyses, we were unable to resolve some of the discrepancies between the data sets. We hypothesized that by applying the TIS method of multiparallel sequencing to a gene-deletion library we would be able to rapidly validate the precise position of every insertion within the library and resolve some of the discrepancies observed. Thus, to test our hypothesis, we pooled the Keio collection of mutants and used the TIS methodology to identify the position of the selection cassette throughout the genome. We present this method, (Linear Insertion) LI-detector, as a tool for rapidly validating any ordered library.

## RESULTS

**Identification of the kanamycin resistance cassette insertion sites within the Keio library.** During the construction of the Keio library, each annotated gene of *Escherichia coli* strain BW25113 was targeted for replacement with a kanamycin resistance cassette. The intention was to create a single gene-deletion mutant with the start codon and seven terminal codons sandwiching the kanamycin resistance cassette (Fig. 1B) (18). To identify the precise insertion sites of the kanamycin resistance cassette within the Keio library, we pooled the mutants and sequenced the resistance cassette-genomic DNA (gDNA) junctions following an amended transposon insertion sequencing protocol (Fig. 1 and Fig. S1). We obtained ~1.6 million reads (accession no. PRJEB37909). Following confirmation and removal of the kanamycin resistance cassette sequence and removal of short or poor-quality reads, 881,943 reads were successfully mapped to the *E. coli* BW25113 reference genome (Table 1; accession no. CP009273.1). During data processing, the largest number of reads were discarded between the Ktag1 and Ktag2 pattern matching steps (Table 1): 'Ktag1' corresponds with the sequence of the K2F forward primer, while 'Ktag2' corresponds with the sequence of the kanamycin resistance cassette immediately downstream from the K2F primer binding site (Fig. S1). The disparity in the number of reads between these two steps is likely

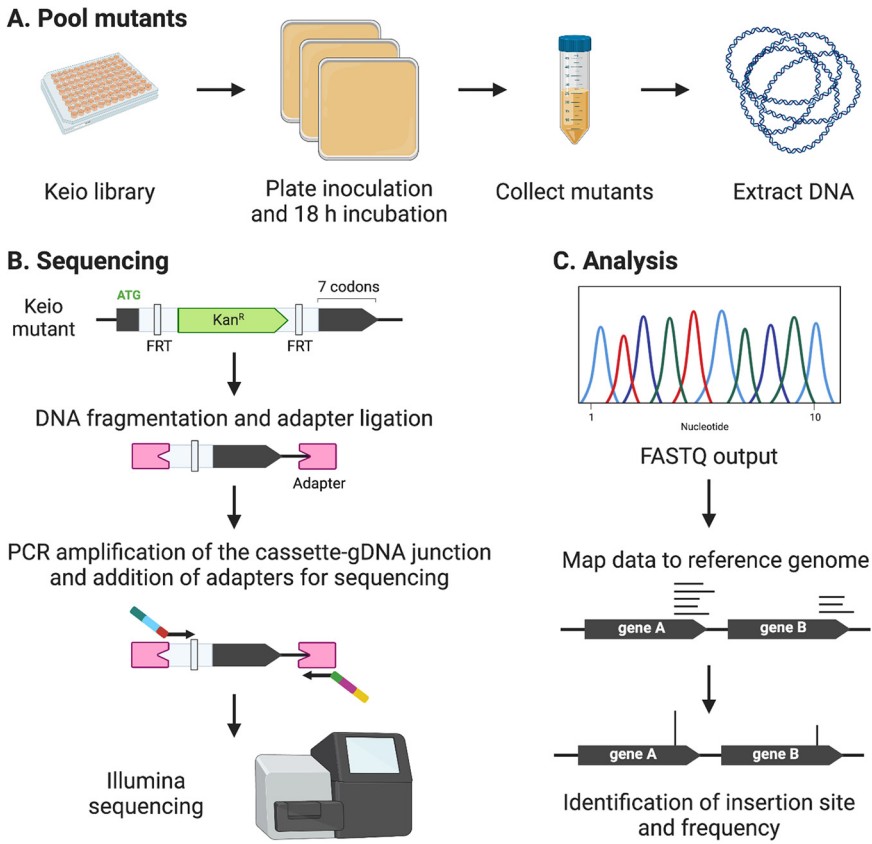

**FIG 1** Sequencing the Keio library. (A) Mutants of the Keio library are stocked in 96-well plates. Each 96-well plate was inoculated onto an LB agar plate and cultures incubated overnight. Colonies were pooled and genomic DNA was extracted from the pool of mutants. (B) The Keio library of mutants was constructed by individually targeting each gene for replacement by the kanamycin resistance cassette amplified from pKD13. The resulting mutant retained a start codon and its terminal 7 codons (inclusive of a stop codon) with the central coding sequence replaced by the resistance cassette. To sequence the Keio library, genomic DNA was extracted from the pooled mutants and fragmented. An adapter was ligated to the fragmented ends, and the cassette-gDNA junction was amplified using primers specific for the 3′ end of the kanamycin phosphotransferase gene and the adapter. (C) Output FASTQ data was mapped to a reference genome, and the first coordinate of each mapped read was counted as the kanamycin resistance cassette insertion site. FRT, Flp recognition target.

a result of mispriming of the K2F forward primer and amplification of off-target DNA; however, data can also be lost at this step if there are single nucleotide polymorphisms (SNPs) in the cassette sequence resulting in barcode mismatch, discussed later. Following read processing, 881,943 mapped reads resulted in the identification of 4,176 kanamycin resistance cassette insertion loci. The mean number of reads per insertion site was 211, while the median number of reads per insertion site was 177. To conservatively determine the presence of mutants within the library, we filtered our sequencing data to remove insertion sites represented by 3 or fewer reads, leaving 4,054 mapped unique insertion sites. Using the *E. coli* BW25113 genome as a reference (CP009273.1) (19), 34 insertion sites were identified in intergenic regions and 4,020 insertion sites were identified within a protein-coding sequence (CDS).

Using the gene names and annotation boundaries of *E. coli* BW25113, we next compared our insertion data with the reported mutants of the Keio library. A gene was con-

**TABLE 1** Number of reads surviving each stage of processing

| No. of barcoded reads | K tag 1 match | K tag 2 match | Reads > 20 bp and passing QC filter | Total mapped reads |
|---|---|---|---|---|
| 1,670,483 | 1,629,609 (97.5%) | 1,076,508 (66.06%) | 1,057,274 (98.21%) | 881,943 (83.42%) |

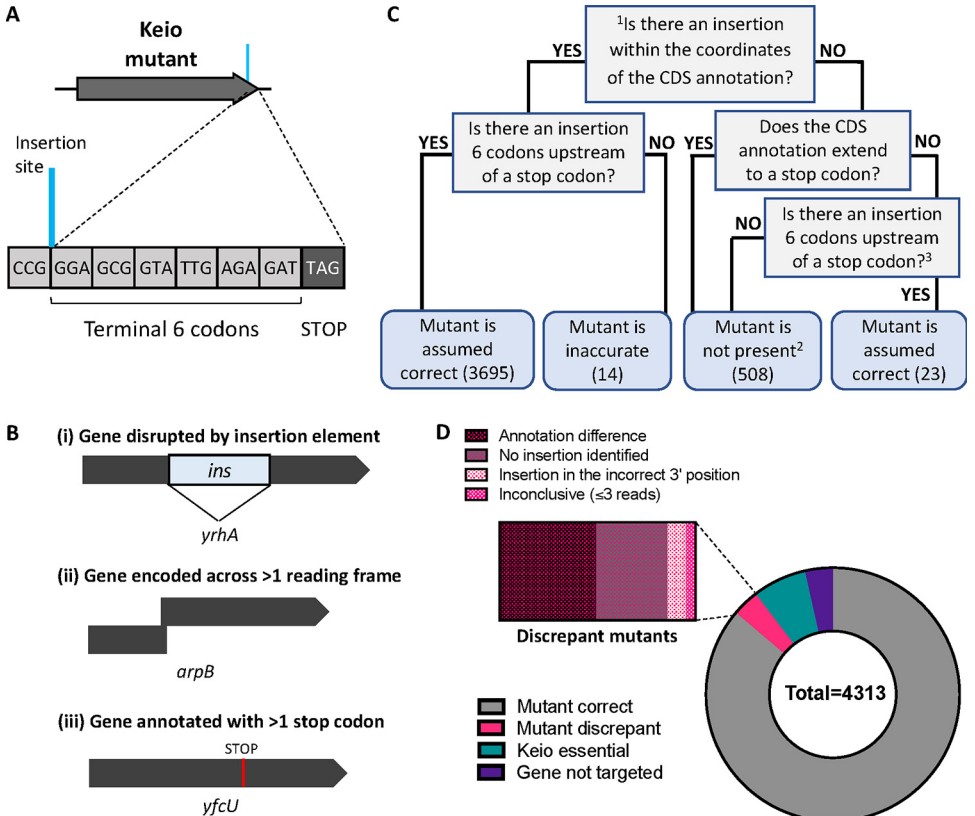

**FIG 2** Identification of the kanamycin resistance cassette insertion sites. (A) Schematic of the correctly located insertion site of a Keio mutant. (B) Examples of genes encoded across >1 reading frame, which were excluded from our analysis. (C) Decision chart for classifying mutants. Caveats: [1]Genes encoded across >1 reading frame were discounted from our analysis (73). [2]The gene could be essential (296); the gene could be a recent annotation and was not targeted for deletion (151); an insertion was detected but had ≤3 reads and was filtered from our analysis (7); or no insertion was detected for an expected mutant (54). [3]Reference genome annotation is incomplete, but the mutant appears correct (23). (D) The presence and status of the corresponding Keio mutant for every annotated gene in *Escherichia coli* BW25113. The kanamycin resistance cassette was correctly positioned within the majority of genes (3,718 genes, gray). For a subset of genes, there were discrepancies with the identified location of the cassette and the gene annotation (148 genes, magenta). There are 296 annotated genes in the BW25113 reference genome reported to be essential during construction of the Keio library (teal) that had no insertion and 151 genes annotated in the BW25113 reference genome that were not targeted for deletion during construction of the Keio library (purple).

sidered correctly disrupted if a cassette insertion site was identified 6 codons upstream from the first stop codon of the targeted open reading frame (ORF) (Fig. 2A). We identified 3,718 genes with a correctly located kanamycin resistance cassette (Fig. 2C and D; Data set S1). We found 23 instances where a CDS is disrupted at the correct position but due to a genome annotation error in the *E. coli* BW25113 reference genome, where the CDS annotation prematurely ends before a stop codon, the identified insertion site appears outside of the annotated CDS coordinates (Data set S1). We identified a nearby but inaccurate (defined as within 3 codons of the correct position) insertion site in 14 genes (Fig. 2C and D). We failed to identify an insertion site for 61 genes, 7 of which had an insertion site represented by ≤3 sequence reads and had therefore been filtered by our analysis (discussed later). For 73 genes the annotation of the reference genome differs from their coordinates given within the Keio library. These genes fall into three categories: (i) genes disrupted by insertion elements; (ii) genes that are encoded across more than one ORF; and (iii) genes that are annotated with more than one stop codon (Fig. 2B).

There are an additional 151 genes annotated in the *E. coli* BW25113 reference genome sequence that were unannotated at the time of construction of the Keio library,

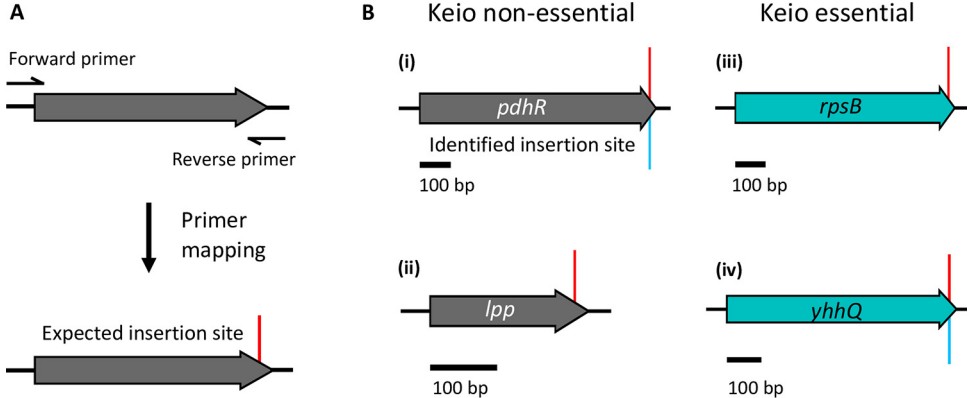

**FIG 3** Comparison of expected and constructed mutants. (A) Primers for construction of a Keio mutant; mapping of the reverse primer to the reference genome reveals the expected insertion site (red). (B) Comparison of expected insertion sites (red) with identified insertion sites (blue). (i) A correctly disrupted gene; (ii) a gene reported to be deleted but with no identified insertion; (iii) a gene reported as essential as the mutant could not be constructed; (iv) a gene reported as essential but with a correctly positioned resistance cassette.

and as such, the corresponding mutants were not reported during the initial construction of the Keio library (13). However, we identified that six of the more recently annotated genes (*yciZ*, *yoaI*, *yegZ*, *ypdJ*, *ypjK*, and *ldrD*) appear to be disrupted in a manner consistent with the Keio mutants (13). Surprisingly, we identified two genes (*yhhQ* and *tnaB*) that were reported to be essential in the initial Keio library paper but are correctly disrupted in our library. As these mutants have been constructed, and our previous TraDIS data supports the notion that these genes are not required for viability, we conclude that these genes are not essential (3). Lastly, the disparity in numbers between the total number of intragenic insertions detected and the total number of mutants can be attributed to the presence of 2 copies of each mutant: for some mutants, we observed both a "correct" copy and an "inaccurate" copy. A comprehensive list of the status of each mutant, with comments, can be found in Data set S1 and the processed insertion data can be viewed at our online browser (https://tradis-vault.qfab.org/).

**Comparison of targeted and identified cassette insertion sites.** We failed to identify the presence of a kanamycin resistance cassette within 61 genes that were targeted for disruption, were predicted to be nonessential by Baba et al. (13), and are thus expected to be present in the library. At the time of construction of the Keio library there was no sequenced genome available for the *E. coli* BW25113 strain, instead closely related strains *E. coli* MG1655 and W3110 were used as references for primer design (Fig. S2A) (13, 20–22). We hypothesized that the failure to detect a mutant might be due to differences in the nucleotide sequences of BW25113 and the strains used for annotation, which would result in primer mistargeting. To explore this, we mapped the 3′ primer sequences (used to construct the mutants) to the *E. coli* BW25113 reference genome (Fig. 3A), enabling a comparison between expected and identified insertion sites (Fig. 3B). Of the initial 4,288 primer sequences, 4,284 mapped successfully, corresponding with 4,276 unique insertion sites. The difference between the number of successfully mapped primers and identified insertion sites was due to more than one primer mapping to the same site. For example, the primer to construct the *stfR* mutant also has homology to the 3′ end of *stfQ*. The four primer sequences (targeting *argI*, *eutA*, *lacI*, and *ygcR*) that did not map to the BW25113 genome were due to differences in the genome annotation and/or sequence at the loci targeted (Supplementary Information). However, despite minor sequence differences, there was still sufficient sequence homology for the *argI*, *lacI*, and *ygcR* mutants to be constructed (Supplementary Information). There was only one mutant (*eutA*) for which we did not identify an insertion in the initial screen and the primer to construct the

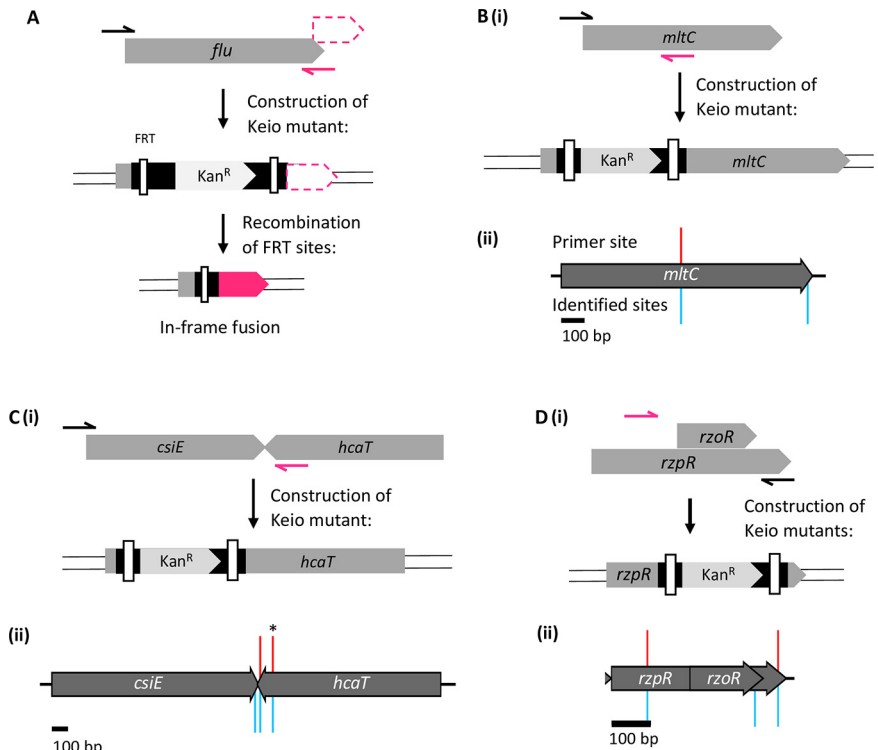

**FIG 4** Identification of the mapped positions of primers used to construct specific Keio mutants (A–D). Construction of the *flu*, *mltC*, *csiE*, and *rzoR* and *rzpR* gene deletion mutants using the primers reported in the original Keio library paper. Incorrectly positioned primers are shown as arrows in magenta, mapped primer sites are shown as red spikes above the gene track, identified insertion sites are shown in blue below. (A) Recombination of the FRT sites within the *flu* mutant will result in an artifact peptide composed of the *flu* start codon, the 34-residue scar left by the cassette, and residues of the in-frame reading frame (magenta). (B) (i) Primers to construct the *mltC* mutant map to the incorrect location (magenta); construction of this mutant would maintain the C-terminal portion of the *mltC* CDS. (ii) Two cassette-gDNA junctions were identified within the *mltC* gene (blue). (C) (i) Primers to construct the *csiE* mutant map to the incorrect location (magenta), construction of this mutant would delete *csiE* and the 3′ end of the neighboring gene *hcaT*. (ii) A cassette-gDNA junctions corresponding with the mislocalized primer position (\*) was identified within the *hcaT* gene (blue), in addition to a correctly position cassette within the *csiE* gene (blue). (D) (i) Primers to construct the *rzoR* and *rzpR* mutants map to the same, incorrect location (magenta). (ii) Sequence data for these loci reveal 2 mapped primer positions (red), and 3 identified cassette insertion sites (blue).

mutant did not map to the *E. coli* BW25113 reference genome. The unique example of *eutA* is discussed later. Overall, the high success rate of primer mapping suggests the failure to identify the 61 mutants cannot be attributed to primer mistargeting.

However, mapping the mutant-construction primers did reveal incorrectly targeted mutants (Fig. 4). Four examples (*flu*, *yedZ*, *mltC*, and *csiE*) of this are presented below. The first of these is the *flu* gene, which encodes Antigen 43 and for which the primer position is incorrect by one nucleotide. The Keio mutants were constructed in such a way that the kanamycin resistance cassette can be excised via recombination of the Flp recombination target (FRT) sites within the cassette (using Flp recombinase) to result in the native gene start codon fused in frame with a 34 amino acid scar and the native terminal 7 codons of the gene. In the example of the *flu* mutant, while the mis-localization of the cassette by 1 nucleotide is relatively minor, this has the unintended consequence of creating an out-of-frame C-terminal fusion mutant when, during removal of the kanamycin resistance cassette, the FRT-sites are recombined (Fig. 4A). The same issue was observed for an additional 7 genes where the cassette is out of frame by 1–2 nucleotides (Data set S1). In each case, excision of the cassette will result in a fusion protein spliced from two reading frames. In this scenario, it is possible that the synthesis of an artifact protein could have unintended effects. The most significant

example of this is the *yedZ* mutant, where excision of the kanamycin resistance cassette will theoretically result in a fused peptide comprised of the *yedZ* start codon, the 34-residue cassette scar peptide and a 51-amino acid C-terminal peptide (Fig. S3).

In some examples, the primer maps to a substantially distant (>9 nucleotides) and incorrect position within the CDS. An example of this is *mltC*; the downstream reverse primer maps to a position in the middle of the CDS (Fig. 4B), which will result in only partial deletion of the *mltC* gene when the mutant is constructed. However, both the truncated *mltC* mutant and the correct *mltC* deletion mutant appear present within the Keio collection (Fig. 4B). Dense transposon mutagenesis screens have shown that provided there is translational read-through or read-out from the inserted cassette, disruptions to the 5′ end of a CDS can still produce a functional protein (3). Partial gene deletions can be problematic if the CDS is functional and confers an alternative phenotype to the WT allele or complete gene deletion (23). The *mltC* gene encodes an outer membrane-bound lytic murein transglycosylase (24). In the truncated *mltC* mutant the transglycosylase portion of the coding sequence is still intact; however, it is out of frame, and therefore should not be expressed (Supplementary Information). There were at least 4 other instances where the original reported primers would result in the construction of a truncated mutant (Data set S1).

Conversely, the reverse primer to construct the *csiE* mutant corresponds to a nucleotide sequence downstream from the correct target site (Fig. 4C). As a result, deletion of *csiE* will also truncate the converging neighboring gene *hcaT*. Deletion of the 3′ end of *hcaT* would result in the loss of the last (12th) transmembrane (TM) domain and disruption of the penultimate TM domain (Supplementary Information), which might affect the localization, folding, stability, or function of the HcaT protein. Partial deletion of a neighboring gene can account for an aberrant phenotype (25). A *csiE* mutant constructed with the incorrectly localized *csiE* primer site was identified within the pool (Fig. 4C, indicated by an asterisk); however, an insertion site was also identified 6 codons upstream from the *csiE* stop codon (Fig. 4C). As such, both the correct and incorrect *csiE* mutants appear present within the library. Finally, the gene *rzoR* is nested within *rzpR*. However, the primers designed to construct each mutant are identical and do not correspond with the appropriate binding sites to correctly construct either mutant (Fig. 4D). It is impossible to distinguish between each respective mutant from the insertion data alone; however, a cassette was also identified 6 codons upstream from the *rzoR* stop codon. Interestingly, the presence of a correctly positioned insertion for *rzoR*, *csiE*, and *mltC* indicates that the library was updated post publication to address errors.

**Validation of Keio mutants by PCR and Sanger sequencing.** There were 61 genes for which we did not detect a cassette insertion site. First, we reevaluated our insertion data. For 7 of these genes (*rnt*, *pykF*, *guaA*, *rlmE*, *rtcA*, *atpC*, and *yccA*) an insertion site was identified in the correct location; however, there were ≤ 3 reads that matched the insertion site and therefore these did not meet our stringency threshold in our initial analysis. Two of these genes, *rnt* and *guaA*, were reported to be essential within a dense transposon library data set (3). As such, the validity of these mutants is inconclusive as the insertion site was in the correct location but with significantly fewer reads than the median number (177). One possible explanation for this is that the glycerol stock of each of these mutants is a mixed population and the correct mutant is in low abundance, and the sequencing method is sensitive enough to identify the correct mutant among the noise. Alternate explanations for a disproportionately low read count include if there was a plating error or if mutants have a slow growth phenotype and form pin-prick colonies. However, we revisited the raw data, this time extracting the sequence data by pattern matching for the expected gene rather than the kanamycin cassette. We identified mutations within the kanamycin cassette sequence upstream of four genes (*rtcA*, *atpC*, *pykF*, and *rlmE*), resulting in these data being filtered during the kanamycin cassette pattern matching steps ('ktag1' and 'ktag2').

Of the remaining 54 genes where no insertion was detected, with the exception of *eutA*, the primer to construct each of these mutants correctly maps to the expected

location within the reference genome. Therefore, to check whether these isolates are indeed incorrect, one copy of each mutant was checked by PCR to map the genomic position of the kanamycin resistance cassette. Briefly, a single colony was isolated and the cassette-gDNA junction was amplified using a primer specific for the kanamycin resistance cassette and a reverse degenerate primer, using a protocol adapted from Levano-Garcia et al. (26). The amplified product was sequenced using Sanger sequencing and a nested primer specific for the cassette. We were unable to check nine isolates as our lab stock of the Keio collection was obtained in 2007 and only includes plates 1–90; more recent versions of the library include plates 91–94. Another explanation for the failure to identify an insertion site is if the mutant failed to grow. In our initial screen, we only identified 4 mutants that did not grow (Table S1). However, during purification for colony PCR, one mutant (*araE*) did not grow upon re-streaking and we propose this isolate was missed in the initial LI-detector screen. For one isolate, *gatA*, although the gene was targeted for deletion and the primer for construction of the mutant maps to the correct location, we did not identify this mutant in our screen. On closer inspection, this gene is not reported as essential in the original paper, but we could not find a record of the *gatA* mutant within 3 independent databases that describe the well positions of each mutant, despite searching for the gene name ('*gatA*'), the ECK number (ECK2087), or b number (b2094). Therefore, we conclude that although this gene was targeted for deletion, this mutant is not present within the Keio library. Of the 43 isolates we were able to validate, 10 were correctly disrupted, but due to nucleotide differences in either the kanamycin resistance cassette or the genome, reads for these data were either binned during processing or did not map to the reference genome (Data set S2; Supplementary Information). Of note, when the position of the kanamycin resistance cassette within the *eutA* mutant was validated by PCR and Sanger sequencing, the cassette was found in the correct locus and is immediately upstream of *eutB*. Both *E. coli* MG1655 and the sequenced *E. coli* BW25113 genome contain a CPZ-55 cryptic prophage inserted into the ethanolamine biosynthesis operon between genes *eutA* and *eutB* (19), which is absent in *E. coli* W3110. As the primer to construct the *eutA* mutant was designed for the *eutAB* junction of W3110, not the neighboring prophage present in *E. coli* BW25113 (Fig. S2B), the primer and sequence data did not map to the BW25113 reference genome.

We identified the kanamycin resistance cassette elsewhere in the genome for the remaining 33 isolates. Three of these (*ybeY*, *dnaT*, and *thyA*) were reported to be essential by TraDIS analysis (3), therefore for these isolates, we checked the position of the kanamycin resistance cassette in the second copy of each mutant. The second copy of *ybeY* did not grow, while the cassette was also incorrect in the additional *dnaT* and *thyA* isolates. We propose that these three genes are essential (Data set S2).

For the 33 isolates where the kanamycin resistance cassette was identified elsewhere in the genome, we noted that the cassette was not randomly distributed throughout the genome, but appeared 6 codons before a stop codon, consistent with Keio mutants. We found that 25 of the misidentified mutants corresponded with a mutant in a neighboring well, 4 of the mutants corresponded with a mutant elsewhere within the same plate, while the kanamycin resistance cassette insertion site of 4 mutants (*ypdI*, *yiaD*, *thyA*, and *oxyR*) correspond with mutants elsewhere in the library (Data set S2). It is therefore likely that these wells were either incorrectly stocked or have been contaminated during the replication of the library.

At least one of the mutants we identified as incorrect (*oxyR*) has been reported previously as an impure mutant (27, 28). To test whether the incorrect mutants in our library are representative of the Keio mutants in other labs, we reviewed the phenotypic profiles of the incorrect mutants using a publicly available phenotypic database for the Keio library (29). The work by Nichols et al. (29) screened the Keio library under a range of different stresses and recorded the resulting colony sizes of each mutant. Using this data, they were able to assign quantitative fitness scores for each gene under each condition. This data can be used to assign a function to genes of unknown function as genes

involved in the same process or pathway have similar phenotypes and therefore have a high correlation score. Therefore, we compared the phenotypic profile of each incorrect mutant with its positive counterpart, for example, the insertion site for the 'hns' mutant was identified within *hupA* (Data set S2), as such, you might expect the 'hns' mutant to behave as the *hupA* mutant. Although we only validated one copy of each mutant clone by PCR, the data presented by Nichols et al. (29) is a combined data set that merges data from both mutant clones; entries with a low correlation coefficient between clones were discarded, therefore we assume that the clones are comparable for all remaining mutants. However, the phenotypic comparison revealed that none of the expected mutants correlated highly ($>$0.6) with the identified mutants (Fig. S4), and only 2 comparisons (*pgaD* with *ycfL*, 0.56; and *kch* with *oppD*, 0.46), had a correlation coefficient greater than 0.4, suggesting the identified contaminations are specific to our library.

## DISCUSSION

The construction of the Keio library was a laborious undertaking and an impressive feat, made all the more remarkable by the fact that despite the discrepancies we identified here and previously, these are very few in number relative to the whole library. Nevertheless, in specific examples highlighted here, some issues remain. Altogether, our method only missed 10 correctly constructed mutants out of a total of 3,728 confirmed isolates. This was a consequence of nucleotide polymorphisms, insertions, or deletions within the kanamycin cassette or the genomic DNA of the mutant resulting in the reads for these data being discarded during processing. Therefore, LI-detector had a success rate of 99.7% for identifying the correct kanamycin cassette position. This data set provides a benchmark for the purity of the Keio mutants and a screening method for mapping the position of an antibiotic resistance cassette in any ordered library. A limitation of this method is that it only informs whether a mutant is present within the total pool of mutants, the nature of the method cannot distinguish whether the mutant of a given well is correct; further validation checks are required to test this. One caveat to consider is that the presence of a correctly positioned kanamycin resistance cassette is not sufficient evidence for the correct construction of a mutant, as previously observed for *yciM*, *yhcB*, *hda*, and *holD* mutants (15, 30). In each of these examples, the kanamycin cassette is in the correct location, but secondary mutations have been reported elsewhere in the genome. Similarly, Yamamoto et al. (14) have reported gene duplication events in a handful of Keio mutants. As such, when working with individual isolates from any ordered library it is recommended to check the mutant by PCR and Sanger sequencing, and when found to be correct it should be transduced into a clean background before use (31).

A noteworthy point is the identification of 73 genes with annotation differences between the *E. coli* BW25113 reference genome and the coordinates used for the initial construction of the Keio library. This highlights the importance of the annotation quality of the reference genome for high-throughput whole-genome screens. Within the *E. coli* BW25113 reference genome annotation (CP009273.1), pseudogenes are annotated with the CDS extending beyond premature stop codons. This can be problematic for two reasons: (1) computational approaches to identify an insertion at a defined location may be confounded, and (2) in some cases the truncated CDSs are still expressed and functional, and therefore not 'pseudogenes.' Additionally, the complexity of the genome is not always reflected in standard genome annotation. For example, the gene *dnaX* encodes 2 subunits of DNA polymerase III: $\tau$ and $\gamma$ expressed from the same locus in equal proportions; subunit $\tau$ is expressed by standard translation, while $\gamma$ is expressed when a -1 nt frameshift occurs at the A_AAA_AAG consensus sequence (32–35). The capacity to encode 2 different protein subunits depending upon ribosomal processivity is not evident from standard gene annotation. This is pertinent for the Keio collection, as some coding sequences in the *E. coli* BW25113 genome have been annotated across multiple coding sequences such as *arpB*, which contains the same frameshift consensus sequence as *dnaX* but, unlike *dnaX*, the *arpB* gene was deleted in two parts. Primer

design is therefore of considerable importance when constructing gene-replacement libraries. We highlighted some specific examples within the Keio library for consideration and demonstrate the utility of our method for identifying these errors.

Finally, we have been able to resolve some of the discrepancies raised in an earlier analysis of a TraDIS library in the *E. coli* strain BW25113 (3). We observed mutations in two genes, *yhhQ* and *tnaB*, previously reported as essential, and conclude that both are nonessential, which is consistent with the TraDIS data. Secondly, we identified that the *ybeY*, *thyA*, and *dnaT* mutants, reported as essential in the TraDIS screen, are not stocked within the Keio library and therefore are likely essential. Conversely, we validated the presence of several mutants in the Keio collection that the TraDIS data predicted to be essential (Data set S1). There are several explanations for the false-positive prediction of essential genes in transposon mutagenesis screens, as has been noted by us and others (3, 17). As a metric, insertion frequency can be influenced by other biological factors besides gene essentiality. Notwithstanding, a transposon insertion sequencing screen provides data-rich information that can be enhanced with the development of more refined analysis methods, in particular the use of machine learning methods. Our data highlight that care needs to be taken with any high-throughput screen to understand the limitations of the library construction method, the validity of the library and the limitations of the analyses used and automation bias; especially with ever increasing high-throughput screens used to assign biological function to genes. We conclude that LI-detector is a quick and robust method for curating ordered gene libraries to identify errors ahead of often costly and time-intensive screens.

## MATERIALS AND METHODS

**Strains and culture conditions.** The Keio library and the parent *E. coli* K-12 strain BW25113 have previously been described (18). All isolates were cultured on Lysogeny broth (LB; 10 g tryptone, 5 g yeast extract, 10 g NaCl) supplemented with kanamycin sulfate, 50 $\mu$g/mL (Fluka Biochemika, Switzerland) and Agar Bacteriological No. 1 (Oxoid, Basingstoke, UK), at 37°C.

**Amplicon sequencing.** Keio library isolates were cultured using a multiple point inoculator to inoculate large square plates of solid LB medium and incubated overnight for 18 h at 37°C. Colonies were scraped from LB plates and pooled. Genomic DNA was extracted using a Qiagen QIAamp DNA Blood minikit, according to the manufacturer's specifications. DNA was fragmented using an ultrasonicator (Diagenode, Bioruptor Plus) with the following profile: 13 cycles of 30 s ON, 90 s OFF at low intensity, to an average fragment size of ~300 bp. The kanamycin resistance cassette-gDNA junctions were prepared for sequencing using NEB Ultra I kit (New England Biolabs). Following adapter ligation and treatment with the NEB USER Enzyme, the sample was purified using AMPure XP beads (Beckman Coulter); to remove adapter dimer and select fragments around ~250 bp, the final sample was eluted in 17 $\mu$L EB buffer, 15 $\mu$L of which was transferred to a microcentrifuge tube. Next, cassette-gDNA junctions were enriched by PCR using a forward primer with homology to the kanamycin resistance cassette (K1F 5'-TC GCCTTCTTGACGAGTTCTTCTAATAAGG-3') and a reverse primer with homology to the ligated adapter (A1R 5'-GACTGGAGTTCAGACGTGTGCTCTTCCGATC-3'). The PCR contained 25 $\mu$L 2× Q5 DNA polymerase (NEB), 2.5 $\mu$L K1F (10 $\mu$M), 2.5 $\mu$L A1R (10 $\mu$M), 15 $\mu$L sample and 5 $\mu$L nuclease-free water and was amplified using the following cycle conditions: 98°C, 48 s (98°C, 15 s; 65°C, 30 s; 72°C, 30 s) ×10; 72°C, 1 min; 4°C hold. Samples were purified using AMPure XP beads at a ratio of 1:0.9, respectively. Finally, samples were prepared for sequencing using a custom forward primer K2F (5'-AATGATACGGCG ACCACCGAGATCTACACTCTTTCCCTACACGACGCTCTTCCGATCTNNNNNNAAAGTATAGGAACTTCGAAGCA GCT-3'), which contains the Illumina P5 flow cell and adapter sequences and a custom barcode denoted by 'N.' The barcodes varied in length from 6 to 9 bp and were unique to each primer. The purpose of the barcode is to add nucleotide diversity during the early cycles of sequencing and stagger the introduction of the kanamycin resistance cassette. Without the barcode, sequencing the cassette-gDNA junction on an Illumina platform would result in 100% nucleotide identity during the early sequencing cycles (corresponding with the uniform cassette nucleotide sequence), and would result in the data being classified as 'overclustered' and subsequently binned. The reverse primers for this reaction were the NEBNext Multiplex Oligos for Illumina (NEB; Fig. S1; Supplementary Information). The PCR contained 25 $\mu$L 2× Q5 DNA polymerase (NEB), 2.5 $\mu$L K2.F (10 $\mu$M), 2.5 $\mu$L NEBNext_oligo.R (10 $\mu$M), 15 $\mu$L sample, and 5 $\mu$L nuclease-free water, and was amplified using the following cycle conditions: 98°C, 48 s (98°C, 15 s; 65°C, 30 s; 72°C, 30 s) ×20; 72°C, 1 min; 4°C hold. Samples were quantified by qPCR using a Library Quantification kit for Illumina platforms (Kapa) and sequenced using an Illumina MiSeq. Data are available from the European Nucleotide Archive (accession no. PRJEB37909).

**Sequencing analysis.** The Fastx barcode splitter and trimmer tools, of the Fastx toolkit, were used to assess and trim the sequences (36). Sequence reads were first filtered by their sample barcodes, allowing zero mismatches. Similarity matching of the cassette was done by identifying the first 35 bp of the kanamycin resistance cassette in two parts: 25 bases (5' to 3', corresponding with the PCR2 primer binding site, termed 'Ktag1') were matched allowing for 3 mismatches, trimmed, and then the remaining

10 bases (corresponding with sequenced cassette, termed 'Ktag2') were matched allowing for 1 mismatch, and trimmed (Fig. S1). Sequences less than 20 bases long were removed using Trimmomatic (37). Trimmed filtered sequences were then aligned to the reference genome *E. coli* BW25113 obtained from the NCBI genome repository (CP009273.1). The aligner bwa was used with the mem algorithm (38). Samtools was used for the subsequent conversion from sam to bam files and the sorting and indexing of data (39). The bedtools suite was used to create bed files, which were intersected against the BW25113 reference genome. Data were inspected manually using the Artemis genome browser, after processing for visualization using custom python scripts (40). The data can also be visualized in our online browser, available at https://tradis-vault.qfab.org/apollo/jbrowse/.

**PCR validation of Keio mutants.** A single colony of each mutant was resuspended in 100 $\mu$L of water and boiled at 98°C for 10 min. Cell debris was pelleted by centrifugation and 5 $\mu$L of supernatant was used as a template for PCR. The cassette-genomic DNA junction was amplified using a primer specific for the kanamycin resistance cassette (Kan-down: 5′-CTATCGCCTTCTTGACGAGTTC-3′) and a reverse degenerate primer (HIB17: 5′-CGGAATTCCGGATNGAYKSNGGNTC-3′) that nonspecifically binds to the genome (26). A 'touchdown' thermal profile was used to maximize primer annealing using the following thermal profile: 95°C for 5 min; then 25 cycles: 45 s at 95°C; 45 s at 60°C, decreasing by 1°C each cycle; 2 min at 72°C; followed by 25 cycles: 45 s at 95°C; 45 s at 50°C; 2 min at 72°C; and then held at 4°C. Following purification, the amplified product was sequenced using Sanger sequencing and a nested primer specific for the kanamycin resistance cassette (Kan-down-seq: 5′-CTAATAAGGGGATCTTGAAG TTC-3′).

**Data availability.** The nucleotide sequence data generated in this study is available at the European Nucleotide Archive (ENA) under accession no PRJEB37909. The data can also be visualized in our online browser available at https://tradis-vault.qfab.org/apollo/jbrowse/.

## SUPPLEMENTAL MATERIAL

Supplemental material is available online only.
**SUPPLEMENTAL FILE 1**, PDF file, 0.5 MB.
**SUPPLEMENTAL FILE 2**, XLSX file, 0.2 MB.
**SUPPLEMENTAL FILE 3**, XLSX file, 0.02 MB.

## ACKNOWLEDGMENTS

We thank Mori for his helpful tables and insight into the Keio library.

The project was devised by F.C.M., A.F.C., J.A.C., and I.R.H. F.C.M. pursued initial investigations, S.A.M. extracted the Keio isolates and prepared the samples for sequencing, and E.S. sequenced the samples using the TraDIS-like protocol. E.C.A.G. processed and analyzed the sequencing data. J.A.B., R.S., I.A.W., C.I., and G.B. validated mutants by PCR. Phenotypic analysis of mutants was done by M.B. The manuscript was written and edited by E.C.A.G., F.C.M., A.F.C., J.A.C., M.B., J.A.B., and I.R.H.

We declare that we have no conflict of interest.

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
