## [Reviewer comments · Microbiology Spectrum]

Microbiology Spectrum

LI-detector: a method for curating ordered gene-replacement libraries

Emily Goodall, Faye Morris, Samantha McKeand, Rudi Sullivan, Isabel Warner, Emma Sheehan, Gabriela Boelter, Christopher Icke, Adam Cunningham, Jeff Cole, Manuel Banzhaf, Jack Bryant, and Ian Henderson

Corresponding Author(s): Ian Henderson, University of Queensland

Review Timeline:

Submission Date:	March 4, 2022
Editorial Decision:	April 18, 2022
Revision Received:	June 29, 2022
Accepted:	July 1, 2022

Editor: John Atack

Reviewer(s): Disclosure of reviewer identity is with reference to reviewer comments included in decision letter(s). The following individuals involved in review of your submission have agreed to reveal their identity: emma rachel Holden (Reviewer #1)

Transaction Report:

DOI: <https://doi.org/10.1128/spectrum.00833-22>

April 18, 2022

Prof. Ian R Henderson
University of Queensland
St Lucia
Australia

Re: Spectrum00833-22 (TraDIS-validate: a method for curating ordered gene-replacement libraries)

Dear Prof. Ian R Henderson:

Thank you for submitting your manuscript to Microbiology Spectrum. As you will see your paper is very close to acceptance. Please modify the manuscript along the lines I have recommended. As these revisions are quite minor, I expect that you should be able to turn in the revised paper in less than 30 days, if not sooner. If your manuscript was reviewed, you will find the reviewers' comments below.

The reviewers have raised several important points that need addressing before this is suitable for Spectrum, in particular those addressing the use of 'TRADIS' in the title, and the overall manuscript length

When submitting the revised version of your paper, please provide (1) point-by-point responses to the issues I raised in your cover letter, and (2) a PDF file that indicates the changes from the original submission (by highlighting or underlining the changes) as file type "Marked Up Manuscript - For Review Only". Please use this link to submit your revised manuscript. Detailed instructions on submitting your revised paper are below.

Link Not Available

Sincerely,

John Atack

Reviewer comments:

Reviewer #1 (Comments for the Author):

This manuscript addresses a very important problem with the Keio collection, in that we know not all of the individual knockout mutants are correct and it is laborious and time-consuming to sequence all the mutants we want to use. I think there is definitely a need for this work and it would be extremely useful to be able to check if the Keio mutants I'm using are correct.

However, I have a few strong feelings about this work that need addressing before it can be published. The first is with the title. I am extremely uncomfortable with calling this work 'TraDIS-validate' without using any transposon directed insertion site sequencing, or even the same BioTraDIS or AlbaTraDIS pipelines used to analyse TraDIS data. The only connection to TraDIS I can see is where the authors compare the gene essentiality findings of the Keio collection with that of their previous TraDIS investigations in BW25113. Pooling an ordered library and sequencing from the kanamycin cassette outwards is definitely similar to the TraDIS protocol, and mentioning this at this point is crucial, but to call the entire approach TraDIS-validate seems disingenuous. I don't believe the tools created to validate ordered libraries can be called TraDIS-validate where no transposon directed insertion site sequencing has occurred. When I first read the title, I thought this work would present a new tool to be incorporated into the TraDIS pipeline, rather than using existing tools for a different use. If this manuscript was renamed to remove the mention of TraDIS, I believe it would better represent a tool to analyse ordered libraries.

Secondly, this manuscript is extremely long for what it is. I can see this being a very thorough thesis chapter, but is excessive for a manuscript. The first section of the results (lines 96-105) repeats the methods and is unnecessary. Detailing the extent of

contamination in the authors' own Keio collection (lines 313-326) is unnecessary, where it would suffice to simply say that this method was able to identify where errors resulted from contamination from neighbouring wells. Thoughts on how best to annotate a genome (line 377) are outside the remit of this work. Overall, it feels like the authors have provided an in depth analysis in the manuscript for every single gene that did not perfectly match the Keio mutant as expected. It would be preferable from a reader's perspective to include gene-by-gene analysis for this in the supplementary materials, and discuss only a few of these in the manuscript.

Overall, I think there is a need for a manuscript and database to validate the Keio collection, I think there is a need for tools to double-check ordered libraries, I believe the execution of the science is sound, but the manuscript does not meet the goals it sets out to achieve. Mentioning TraDIS in the title gives a false view of the manuscript's contents and extensive rewriting is needed to make this a publishable manuscript.

Reviewer #2 (Comments for the Author):

In this manuscript, Goodall et al. describe a validation method for TraDIS gene replacement libraries as well as a benchmarking of the method by applying it to the Keio knock-out library of E. coli K-12. Their PCR-based validation method is both novel and beneficial in that they not only confirm the presence of the expected insertion mutant but also identify the precise location of genomic insertion and in doing so can also confirm if the insertion took place in the expected location and if a frame-shift mutation could result when the cassette is excised. The authors describe in detail their efforts to validate each mutant, their findings in terms of likely essential genes (or genes that were previously thought to be essential, but mutants were found within this library), insertions into an unexpected or incorrect location, and identification of the mutant from another well in the plate, indicating the expected mutant is not present due to crossover/contamination. The work was well described in this manuscript.

Preparing Revision Guidelines

- point-by-point responses to the issues I raised in your cover letter
- Upload a compare copy of the manuscript (without figures) as a "Marked-Up Manuscript" file.
- Each figure must be uploaded as a separate file, and any multipanel figures must be assembled into one file.
- Manuscript: A .DOC version of the revised manuscript
- Figures: Editable, high-resolution, individual figure files are required at revision, TIFF or EPS files are preferred

Please return the manuscript within 60 days; if you cannot complete the modification within this time period, please contact me. If you do not wish to modify the manuscript and prefer to submit it to another journal, please notify me of your decision immediately so that the manuscript may be formally withdrawn from consideration by Microbiology Spectrum.

17 February 2022

Dear Dr. Atack,

Please find our responses to referees comments

Referee 1:

A. *I am extremely uncomfortable with calling this work 'TraDIS-validate'. We agree with the referee and have changed the title to **LI-detector: a method for curating ordered gene-replacement libraries**. It is worth noting that the name "TraDIS" is itself disingenuous, as no transposon is used.*

B. *Secondly, this manuscript is extremely long for what it is. We have edited and substantially reduced the verbiage in the manuscript*

Reviewer 2:

We thank the referee and note no corrections were suggested

July 1, 2022

Prof. Ian R Henderson
University of Queensland
Institute for Molecular Bioscience
306 Carmody Road
St Lucia, QLD 4068
Australia

Re: Spectrum00833-22R1 (LI-detector: a method for curating ordered gene-replacement libraries)

Dear Prof. Ian R Henderson:

Your manuscript has been accepted, and I am forwarding it to the ASM Journals Department for publication. You will be notified when your proofs are ready to be viewed.

Sincerely,

John Atack
Editor, Microbiology Spectrum

Journals Department
Supplemental Material: Accept
Supplemental Dataset: Accept
Supplemental Dataset: Accept